# Echocardiography Assessment of Cardiac Function in Adults Living with HIV: A Speckle Tracking Study in the Era of Antiretroviral Therapy

**DOI:** 10.3390/jcm11133792

**Published:** 2022-06-30

**Authors:** Oana Mirea, Mirela Manescu, Sorina Iordache, Andreea Marcu, Ionut Donoiu, Octavian Istratoaie, Florentina Dumitrescu, Constantin Militaru

**Affiliations:** 1Department of Cardiology, University of Medicine and Pharmacy Craiova, 200349 Craiova, Romania; oana.munteanu@umfcv.ro (O.M.); manescumirela@yahoo.com (M.M.); sorina.iordache93@gmail.com (S.I.); octavian.istratoaie@umfcv.ro (O.I.); constantin.militaru@umfcv.ro (C.M.); 2Department of Infectious Diseases, University of Medicine and Pharmacy of Craiova, 200349 Craiova, Romania; busuandreea01@gmail.com (A.M.); florentina.dumitrescu@umfcv.ro (F.D.)

**Keywords:** acquired immunodeficiency syndrome, diastolic dysfunction, strain

## Abstract

Adults living with HIV (human immunodeficiency virus) infection (ALHIV) have high rates of cardiovascular events. New approaches are needed to detect subclinical cardiac dysfunction. We used conventional and speckle tracking echocardiography to investigate whether ALHIV display latent cardiac dysfunction. We analyzed 85 young subjects with HIV infection and free from cardiovascular risk factors (31 ± 4 years) and 80 matched healthy volunteers. We measured left ventricular (LV) layered global longitudinal strain, circumferential strain, peak longitudinal strain in the reservoir and contraction phases of the left atrium (LASr respectively LASct). In the HIV group, LV ejection fraction and s’ TDI (tissue doppler imaging) were slightly lower but still in the normal ranges. Layered longitudinal strain showed no significant difference, whereas circumferential global strain was significantly lower in the HIV group (−20.3 ± 3.9 vs. −22.3 ± 3.0, *p* < 0.001). LASr (34.3% ± 7.3% vs. 38.0% ± 6.9%, *p* < 0.001) was also lower in ALHIV and multivariate analysis showed that age (β = −0.737, *p* = 0.01) and infection duration (β = −0.221, *p* = 0.02) were independently associated with LASr. In the absence of cardiovascular risk factors, adults living with HIV display normal LV systolic function. Left atrial reservoir strain, is, however, decreased and suggests early diastolic dysfunction.

## 1. Introduction

In 2020, approximately 37 million adults were living with human immunodeficiency virus (HIV) infection, and 75% were receiving antiviral therapy. The development of targeted therapy protocols such as antiretroviral therapy (ART) and the increasing accessibility to treatment led to a steep decrease in mortality among adults living with HIV (ALHIV) [1].

The effective suppression of viral replication significantly lowered the prevalence of acute opportunistic infections and oncologic complications, shifting the clinical spectrum of HIV infection towards chronic, more subtle conditions of the heart, liver and kidney. The relationship between HIV and cardiac disease was early documented and exceedingly studied. HIV infection is associated with an adverse metabolic and cardiovascular risk profile [2], and the presence of overt cardiac dysfunction markedly increases the risk of death among ALHIV [3].

Despite effective treatment, there is still evidence that the rate of acute cardiac events remains higher in ALHIV, beyond that explained by recognized risk factors [4,5]. Thus, early recognition of cardiac impairment is fundamental since timely medical interventions could prevent or delay irreversible cardiac dysfunction.

Speckle tracking echocardiography has become a valuable clinical tool for the quantification of myocardial function [6]. Global longitudinal strain was shown to be a sensitive marker of LV impairment, with diagnostic and prognostic information even in the presence of normal ejection fraction (EF). Moreover, left atrial (LA) strain is an established predictor of future cardiac events in the general population [7].

There have been relatively few studies addressing cardiac abnormalities detectable by speckle tracking echocardiography among adults infected with HIV in the modern era of antiretroviral therapy. Thus, we investigated using a multiparametric approach whether speckle tracking echocardiography-derived indices of both ventricles and the left atrium can detect early cardiac dysfunction in young patients with HIV infection, free from conventional risk factors.

## 2. Materials and Methods

### 2.1. Study Population

We prospectively recruited 85 patients (mean age 31 ± 4; 45 males) with acquired HIV infection between June 2020 and June 2021 from a single center. Inclusion criteria were (1) age between 18 and 45 years old; (2) presence of sinus rhythm and (3) ability to sign an informed consent. Exclusion criteria were (1) history of cardiomyopathy, coronary artery disease or valvular heart disease; (2) presence of diabetes mellitus; (3) history of arterial pressure above 140 mmHg or (4) echocardiographic images unsuitable for quantification. The control group was selected among 80 age- and gender-matched healthy volunteers with low risk of HIV infection and no history of any cardiovascular disease and with normal findings on clinical examination, electrocardiography and echocardiography.

Blood pressure, heart rate and anthropometric information including age, weight and body surface area (BSA) were collected at the time of study. Available clinical data and laboratory results as well as the treatment protocols (for the study group) were obtained from the medical records of the patient.

Written informed consent was obtained from each subject, and the study was approved by the ethics committee of our institution.

### 2.2. Conventional Echocardiography

Transthoracic echocardiographic images were recorded using a commercially available ultrasound system (Vivid iQ, equipped with a 3.5 MHz transducer, GE Vingmed Ultrasound, Horten, Norway). Echocardiography was performed and analyzed offline by two experienced cardiologists (M.M. and O.M.). Subjects were examined at rest, positioned in the left lateral decubitus. A minimum of three cardiac cycles/image were acquired with a mean frame rate of 60 fps.

All echocardiographic measurements were obtained in accordance with the current recommendations for chamber quantification [8]. Left ventricular wall thickness and cavity diameters were measured from the parasternal long axis view. To assess LV systolic function, we calculated ejection fraction (LVEF) using Simpson biplane method from four and two chamber views.

Diastolic function was estimated by measuring the peak early (E) and late (A) transmitral diastolic peak flow velocities (m/s), the E/A ratio and E wave deceleration time (DTE). Peak systolic (s’), early diastolic (e’) and late diastolic (a’) mitral annular velocities were obtained by averaging the values recorded at the septal and lateral positions. To assess LV filling pressures, the E/e’ ratio was calculated.

Left atrium volume was obtained from the four-chamber view at end-systole and end-diastole. The right atrium (RA) volume as well as RV end-diastolic area (EDA) and end-systolic area (ESA) and the fractional area change (FAC) were measured from the modified 4-chamber view. The RV function was estimated from tricuspid annular plane systolic excursion (TAPSE) measurements and the TDI systolic values of the free lateral wall. Pulmonary arterial systolic pressure (PASP) was estimated by Doppler echocardiography from the systolic right ventricular to the right atrial pressure gradient using the modified Bernoulli equation (4 times the peak tricuspid velocity-squared). In the presence of a nondilated inferior vena cava with >50% collapsibility, right atrial pressure was assumed to be 6 mmHg. When appropriate, parameters were indexed to BSA.

### 2.3. Speckle Tracking Echocardiography

Speckle tracking analysis was performed using EchoPAC version 204 software (GE Vingmed Ultrasound). For all LV, RV and LA measurements, zero strain was defined by the automatic R wave trigger on the ECG. The range ROI was changed if deemed necessary, and the tracking of each segment was visually assessed and rejected when not appropriate. Patients were excluded from the strain analysis if the regional tracking of one or more segments was suboptimal. Aortic valve closure was set on the pulsed wave doppler recording of the LV outflow track.

#### 2.3.1. Left Ventricular Strain

A left ventricular 17-segment model was used. In summary, endocardial border was tracked in 2-, 3-, and 4-chamber views. Global longitudinal strain (GLS) for each layer (endocardium, myocardium and epicardium) was provided automatically (Figure 1A). Peak longitudinal strain (LS) segmental dispersion (PSD) was calculated as the standard deviation of LS segmental time to peak (expressed in milliseconds). Global circumferential strain (GCS) was assessed from the short axis views (mitral valve, papillary muscles and apex) and obtained by averaging the circumferential strain in the three views (Figure 1B).

#### 2.3.2. Right Ventricular Strain

RV global longitudinal strain was obtained from the modified four-chamber view by tracing the endocardium border. For RV global LS, a six-segment model was used. RV free wall strain was calculated by averaging the segmental strain (RV lateral wall basal, mid and apex) (Figure 1C).

#### 2.3.3. Left Atrial Strain

LA strain was obtained using the dedicated tool for LA strain (EchoPac v 204). In summary, three anatomic landmarks were positioned manually (the sides of the mitral valve and the LA roof. LA strain during the reservoir phase (LASr) and LA strain during the contraction phase (LASct) were automatically provided (Figure 1D).

### 2.4. Statistical Analysis

Clinical and echocardiographic measurements are provided as mean ± standard deviation for continuous variables or as absolute number or percentage for categorical variables. Variables were checked to be normally distributed and to have equal variances. To compare data between the predefined study groups, the unpaired t-test was used. Significance was set at a two-tailed probability level of <0.05. Statistical analyses were performed using SPSS version 17.0 (SPSS Inc., Chicago, IL, USA).

## 3. Results

### 3.1. Clinical Data

There were no major differences in anthropometric characteristics between the groups (Table 1). The mean time from diagnosis of HIV infection was 17 ± 8 years (70% parenteral transmission, 30% sexual transmission). The treatment protocol is summarized in Table 1. A total of 15%of the HIV patients were active smokers.

### 3.2. Conventional Echocardiography

LV wall thickness and mass were higher in ALHIV (*p* < 0.01) (Table 2), whereas LVEF was slightly lower in the HIV group, reaching borderline significance (58 ± 6% vs. 60 ± 7%, *p* = 0.06). LA end-systolic and end-diastolic volumes were also similar between the groups (*p* = 0.28 and *p* = 0.34). Eighteen (21%) HIV patients showed diastolic dysfunction grade I. A decrease in e’ and increase in the E/e’ ratio were observed in the HIV group (*p* < 0.01 for both).

Right heart measurements were also similar between the groups (Table 2) except for s’ of the RV free wall, which was lower in ALHIV (11.9 ± 2.3 vs. 13.2 ± 2.0, *p* < 0.001).

### 3.3. Speckle Tracking Echocardiography

The feasibility was 98% for LV strain, 89% for RV strain and 93% for LA strain.

#### 3.3.1. LV Strain Analysis

In all subjects, strain values decreased from endocardium to epicardium. The comparison of LV and LS indexes between the groups is shown in Table 3. No significant differences were observed. PSD was higher in ALHIV, reaching borderline significance (32 ± 10 ms vs. 29 ± 11 ms, *p* = 0.06). GCS was significantly lower in ALHIV (−20 ± 3 vs. −22 ± 3, *p* < 0.001) (Table 3).

#### 3.3.2. RV Strain Analysis

RV global strain and free wall strain were significantly lower in ALHIV (−21.8 ± 3.4 vs. −23.1 ± 3.3, *p* = 0.04 and −25.9 ± 4.1 vs. −27.7 ± 3.8, *p* = 0.02, respectively). Using the univariate logistic regression analysis, parenteral infection (β = −0.527, *p* = 0.032), duration of HIV infection (β = 0.711, *p* = 0.004) and BMI (β = 0.404, *p* = 0.004) were independent predictors of RV global strain.

#### 3.3.3. LA Strain Analysis

The LA reservoir average was 38 ± 7% in the control group and significantly lower (34 ± 7%) in patients with HIV (*p* < 0.001). Univariate logistic regression analysis demonstrated that age (β = −0.737, *p* < 0.001) and years of HIV infection (β = −0.221, *p* < 0.05) were independent predictors of LA strain. LAS during contraction showed no difference between the groups.

## 4. Discussion

The main findings of the present study can be summarized as follows: (a) Young patients with long-lasting HIV infection receiving ART show normal LV systolic function; (b) RV global and free wall LS were lower in ALHIV and (c) LA reservoir strain was significantly reduced among ALHIV.

We prospectively included young patients with long-lasting HIV infection (mean duration of 17 ± 8 years). This allowed us to more accurately study the effect of HIV infection in subjects free from cardiovascular risk factors. Adults living with HIV showed a mild increase in LV wall thickness and subsequent LV mass compared with controls. The association between increased LV mass and high all-cause mortality was previously reported in HIV patients [9,10]. The LV ejection fraction was slightly lower in the HIV group compared with controls, although still in the normal range. Echocardiographic studies have reported a high prevalence of diastolic dysfunction, ranging from 40 to 50%, in ALHIV [9,11,12]. Nevertheless, newer reports found that in individuals without cardiovascular risk factors such as diabetes or hypertension, diastolic dysfunction was almost absent [13]. In our study, the prevalence was also lower (21%), and it could be explained by the low incidence of cardiovascular risk factors in our population.

During the past decade, GLS has consolidated its clinical value in assessing early LV impairment. In our study, endocardial, myocardial and epicardial GLS were similar between the groups, suggesting normal LV function in ALHIV. Our results are difficult to compare with previous studies as patient characteristics and patient numbers vary considerably. However, small studies using two- and three-dimensional strain measurements reported impaired LV deformation in ALHIV, regardless of the ART [14,15]. Similar findings were reported by Cincin et al. in a larger study, including 120 patients with HIV. However, the patients were significantly older compared with our study group and had associated hypertension, dyslipidemia and diabetes [16]. Peak segmental LS dispersion was proposed as an indicator of myocardial heterogeneity and is considered a valuable assessor of arrhythmic risk in various cardiac diseases [17,18]. ALHIV had higher LV peak strain dispersion reaching borderline significance. This is likely due to the higher LV mass that we observed in ALHIV. Additionally, global circumferential strain was significantly lower in ALHIV. An impaired circumferential shortening was shown to have negative impact on LVEF, even in the presence of normal longitudinal shortening [19], and could explain the mild reduction in EF that we observed in the HIV group. Lastly, an impaired GLS was reported to be associated with carotid atherosclerosis in people living with HIV, suggesting possible links between the two conditions [20].

Right ventricular global strain and free wall strain showed lower values in the HIV group compared with controls. While there is no clear pathophysiologic explanation, early studies also reported isolated RV dysfunction in people living with HIV independent from pulmonary hypertension [21,22]. Moreover, an increased pulmonary arterial stiffness was observed in patients with HIV [23].

Left atrial strain was shown to be a powerful predictor of cardiovascular events in subjects free from cardiac disease [7] as well as in subjects with associated pathology [24,25,26]. Moreover, studies advocate that LA strain is a finer measurement than LA volume and could replace the latter in the algorithm for LV diastolic dysfunction assessment [27]. While LA end diastolic and end systolic volumes showed no differences between the groups, LA reservoir strain was significantly lower in the HIV group. This implies that LA strain is a more sensitive marker of early cardiac dysfunction, with changes earlier than structural changes. Interestingly, since LV GLS was shown to be the prime determinant of LA strain [28] and we could not find any significant changes of GLS, it raises the question of a primary abnormality of the LA in HIV patients.

Several studies have related antiretroviral therapy to the development of arterial hypertension [29,30] or dyslipidemia [31], while others found no relationship. The implication of ART for cardiac function was previously studied [32]. In our study, we found no correlation between treatment protocol and LV function. Moreover, on multivariate analysis, the choice of treatment was not significantly associated with cardiac impairment.

The main limitation of this study was the small number of patients with HIV infection. Our population was highly selected, and we included only young asymptomatic subjects without cardiovascular risk factors. Moreover, as only treated patients were included, the role of antiretroviral therapy could not be assessed independent of HIV infection. Lastly, in the absence of complete data, we could not assess the impacts of viral load on cardiac function.

## 5. Conclusions

Young asymptomatic adults with HIV infection and antiviral therapy present with normal left ventricular systolic function. The assessment of LA strain dynamics by speckle tracking echocardiography in HIV patients may be of particular interest in those with no evidence of LA enlargement.

## Figures and Tables

**Figure 1 jcm-11-03792-f001:**
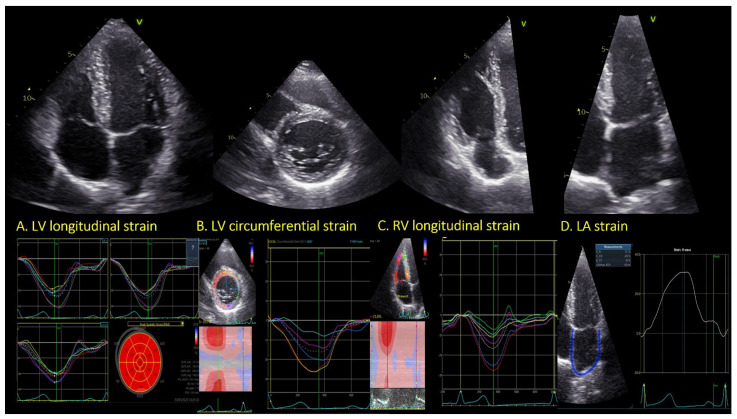
Examples of speckle tracking analysis.

**Table 1 jcm-11-03792-t001:** Anthropometric and clinical measurements.

		HIV Group (*n* = 85)	Controls (*n* = 80)	*p* Value
**Clinical characteristics**				
Age	(years)	31.7 ± 4.3	30.1 ± 8.0	0.12
BSA	(m^2^)	1.81 ± 0.2	1.83 ± 0.2	0.87
Parenteral infection	*n* (%)	60 (70%)	-	-
Sexual infection	*n* (%)	25 (30%)	-	-
Current smokers	*n* (%)	15 (15%)	14 (17%)	0.11
Heart Rate	(bpm)	74 ±10	77 ± 11	0.20
Systolic blood pressure	(mmHg)	118 ± 15	121 ± 11	0.87
Diastolic blood pressure	(mmHg)	68 ± 8	67 ± 7	0.76
**Treatment protocol**				
RTIs	*n* (%)	34 (40%)	-	
RTIs + NNRTIs	*n* (%)	21 (25%)	-	
RTIs + PI	*n* (%)	17 (20%)	-	
RTIs + II	*n* (%)	13 (15%)	-	
**Biologic characteristics**				
LyT CD_4_+	(cells/mm^3^)	559 ± 308	-	
LyT CD_8_+	(cells/mm^3^)	826 ± 430	-	
Creatinine	(mg/dL)	1.0 ± 0.1	-	
Urea	(mg/dL)	45 ± 15	-	
LDL-C	(mg/dL)	118 ± 42	-	
HDL-C	(mg/dL)	70 ± 12	-	
Triglycerides	(mg/dL)	225 ± 183	-	
ALT	(U/I)	41 ± 30	-	
AST	(U/I)	42 ± 33	-	

ALT: alanine aminotransferase; AST: aspartate aminotransferase; BSA: body surface area; DBP: diastolic blood pressure; HDL-C: high-density lipoprotein cholesterol; LDL-C: low-density lipoprotein cholesterol; SBP: systolic blood pressure; RTIs: reverse-transcriptase inhibitors; NNRTIs: non-nucleoside reverse transcriptase inhibitors; PI: protease inhibitors; II: integrase inhibitors.

**Table 2 jcm-11-03792-t002:** Conventional echocardiographic measurements.

		HIV Group (*n* = 85)	Controls (*n* = 80)	*p* Value
LV EDD	(mm)	45.5 ± 5.0	45.9 ± 4.5	0.60
LV ESD	(mm)	31.7 ± 5.2	30.6 ± 4.4	0.16
IVSd	(mm)	9.3 ± 1.6	8.6 ± 1.6	<0.01
PWd	(mm)	9.2 ± 1.6	8.6 ± 1.4	<0.01
LVMi	(g/m^2^)	78.8 ± 19.3	70.7 ± 15.5	<0.01
LV EDVi	(mL/m^2^)	51.1 ± 8.7	48.3 ± 10.2	0.07
LV ESVi	(mL/m^2^)	21.4 ± 4.5	19.3 ± 5.6	0.01
LV EF	(%)	58 ± 6	60 ± 7	0.06
LV TDI s’	(cm/s)	8.0 ± 1.7	9.2 ± 1.8	<0.01
E wave	(cm/s)	74 ± 14	75 ± 14	0.82
A wave	(cm/s)	58 ± 12	53 ± 13	<0.05
E/A ratio	-	1.3 ± 0.4	1.5 ± 0.5	<0.01
DTE	(ms)	174 ± 39	163 ± 26	0.07
LV TDI e’	(cm/s)	12.4 ± 2.6	14.8 ± 2.0	<0.001
LV TDI a’	(cm/s)	8.7 ± 2.4	8.6 ± 2.0	0.84
E/e’	-	6.2 ± 1.4	5.2 ± 1.4	<0.001
LA ESVi	(mL/m^2^)	21 ± 5	22 ± 7	0.20
LA EDVi	(mL/m^2^)	9 ± 3	10 ± 3	0.17
RV EDAi	(cm^2^)	8.7 ± 1.6	8.9 ± 42.5	0.54
RV ESAi	(cm^2^)	4.8 ± 1.1	5.0 ± 1.6	0.24
RV FAC	(%)	45 ± 7	43 ± 8	0.20
RA ESVi	(mL/m^2^)	17.5 ± 4.2	17.1 ± 4.2	0.51
RA EDVi	(mL/m^2^)	9.3 ± 3.1	8.6 ± 2.8	0.15
RV TDI s’	(cm/s)	11.9 ± 2.3	13.2 ± 2.0	<0.001
RV TDI e’	(cm/s)	12.2 ± 2.0	15.1 ± 3.5	<0.001
RV TDI a’	(cm/s)	11.9 ± 3.2	12.2 ± 3.7	0.57
TAPSE	(mm)	21.2 ± 3.5	22.4 ± 3.2	0.04
PAPs	(mmHg)	22.1 ± 4.2	21.6 ± 5.7	0.61

A: pulsed Doppler transmitral peak late diastolic wave; a’: tissue Doppler peak late diastolic wave; DTE: transmitral E wave deceleration time; EDA: end-diastolic area; EDD: end-diastolic diameter; EDVi: end-diastolic volume index; ESA: end-systolic area; EF: ejection fraction; E/A: transmitral peak E/A ratio; E/e’: peak transmitral E/tissue Doppler e’ wave ratio; e’: tissue Doppler peak early diastolic wave; ESVi: end-systolic volume index; IVSd: interventricular septum thickness, diastole; LA: left atrial, myocardial performance index; LV: left ventricular; LVMi: left ventricular mass index; PAPs: systolic pulmonary artery pressure; PWd: postero-lateral wall thickness, diastole; s’: tissue Doppler peak systolic wave; TAPSE, tricuspid annular systolic excursion; TDI: tissue Doppler imaging; RV: right ventricle.

**Table 3 jcm-11-03792-t003:** Speckle tracking-derived strain measurements.

		HIV (*n* = 100)	Controls (*n* = 70)	*p* Value
**Left ventricle**				
GLS endocardial	%	−22.7 ± 2.5	−22.6 ± 3.0	0.82
GLS myocardial	%	−19.7 ± 2.2	−19.7 ± 2.5	0.88
GLS epicardial	%	−17.2 ± 2.0	−17.3 ± 2.4	0.74
PSD	ms	32.2 ± 9.8	28.9 ± 11.4	0.06
GCS	%	−20.4 ± 3.0	−22.3 ± 3.1	0.001
**Right ventricle**				
RV GLS	%	−21.8 ± 3.4	−23.1 ± 3.3	0.04
RV free wall	%	−25.9 ± 4.1	−27.7 ± 3.8	0.02
**Left atrium**				
LAS res	%	34.2 ± 7.3	38.0 ± 6.9	0.001
LAS ct	%	11.2 ± 5.2	11.2 ± 5.2	0.83

GCS: global circumferential strain; GLS: global longitudinal strain; LAS ct: left atrium strain during contraction; LAS res: left atrium reservoir strain; PSD: peak strain dispersion; RV: right ventricle.

## Data Availability

Data available on request with the approval of the University of Medicine and Pharmacy of Craiova.

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
