# Peer review of "Echocardiography Assessment of Cardiac Function in Adults Living with HIV: A Speckle Tracking Study in the Era of Antiretroviral Therapy"

_jcm, 2022, doi:10.3390/jcm11133792_

Round 1

Reviewer 1 Report

Dear author;

I reviewed the article entitled ‘: Echocardiography assessment of cardiac function in subjects  with HIV: a speckle tracking study in the era of antiretroviral  therapy ’. I found the article very interesting and useful for our journal. However, I have some minor comments before the acceptance of article.

--minor comments;

1-There are some grammatical mistakes in the article, hence, I recommend the proof-reading

2- Lastly, please cite this article: PAssessment of pulmonary arterial stiffness in patients with cirrhosis: A prospective cohort study. Öz A, Çınar T, TaÅŸ E, ÇaÄŸan Efe S, Ayça B, KarabaÄŸ T. Echocardiography. 2021 Jan;38(1):57-63. 

Author Response

We would like to thank the editors and reviewers for the detailed review of the manuscript and their helpful comments. We have tried to address point by point the raised concerns.

Reviewer 1

” Dear author,

I reviewed the article entitled ‘: Echocardiography assessment of cardiac function in subjects with HIV: a speckle tracking study in the era of antiretroviral therapy’. I found the article very interesting and useful for our journal. However, I have some minor comments before the acceptance of article.”

--minor comments;

Comment 1-There are some grammatical mistakes in the article, hence, I recommend the proof-reading

Reply to comment 1

Thank you for the observation. The manuscript was thoroughly revised, and corrections have been made.

Comment 2- Lastly, please cite this article: Assessment of pulmonary arterial stiffness in patients with cirrhosis: A prospective cohort study. Öz A, Çınar T, TaÅŸ E, ÇaÄŸan Efe S, Ayça B, KarabaÄŸ T. Echocardiography. 2021 Jan;38(1):57-63. 

Reply to comment 2

We thank the Reviewer for the suggestion. We have indeed addressed the suggested manuscript in the text and included the reference section (reference no. 23).

Reviewer 2 Report

The manuscript, “Echocardiography assessment of cardiac function in subjects with HIV: a speckle tracking study in the era of antiretroviral therapy” is a cases-control study for comprehensive echocardiography in individuals living with HIV. Overall, this study is well conceived and additive to the current literature. Previous work has evaluated cardiac function utilizing circumferential and longitudinal strain predominantly. This work ulitizes GE EchoPac strain and comprehensively evaluates the heart. This is not the first study of this kind, but there is relatedly few to date and I have not found any with EchoPac.

Using Phillips software, Cincin et al reported similar results as cited by the authors. Interestingly, no difference was seen in an US veteran cohort. This could be related to additional cardiovascular risk factors in an aging cohort.  Berg, Christopher, et al. "LEFT ATRIAL STRAIN AND DIASTOLIC DYSFUNCTION AMONGST HIV-POSITIVE INDIVIDUALS: INSIGHTS FROM THE VETERANS AGING COHORT STUDY." Journal of the American College of Cardiology 77.18_Supplement_1 (2021): 1422-1422.

General Comments:

Terminology of “subjects with HIV” should be avoided. Consider adults or people living with HIV and associated abbreviations (PLHIV).

Overall good well measured discussion.

Abstract:

Page 1, Line 25: LV systolic ____. Word missing.

Introduction:

Page 1 line 36: “rather” is wrong word choice.

Page 1 lines 37-38:  references should be added to sentence “The relationship …”

Page 2 lines 48-50: sentence “Moreover, …” Need to specific not in HIV population.

Methods:

Page 2 line 64: Are you reported the gender or the sex?

Page 2 line 95: define PASP.

Results:

Page 5 Lines 168-170: Indentation error.

How was route of infection assessed? Is this accurate and relevant?

Author Response

We would like to thank the editors and reviewers for the detailed review of the manuscript and their helpful comments. We have tried to address point by point the raised concerns.

Reviewer 2

The manuscript, “Echocardiography assessment of cardiac function in subjects with HIV: a speckle tracking study in the era of antiretroviral therapy” is a cases-control study for comprehensive echocardiography in individuals living with HIV. Overall, this study is well conceived and additive to the current literature. Previous work has evaluated cardiac function utilizing circumferential and longitudinal strain predominantly. This work ulitizes GE EchoPac strain and comprehensively evaluates the heart. This is not the first study of this kind, but there is relatedly few to date and I have not found any with EchoPac. Using Phillips software, Cincin et al reported similar results as cited by the authors. Interestingly, no difference was seen in an US veteran cohort. This could be related to additional cardiovascular risk factors in an aging cohort.  Berg, Christopher, et al. "LEFT ATRIAL STRAIN AND DIASTOLIC DYSFUNCTION AMONGST HIV-POSITIVE INDIVIDUALS: INSIGHTS FROM THE VETERANS AGING COHORT STUDY." Journal of the American College of Cardiology 77.18_Supplement_1 (2021): 1422-1422.

General Comments:

Terminology of “subjects with HIV” should be avoided. Consider adults or people living with HIV and associated abbreviations (PLHIV).

Reply to General Comments: We thank the reviewer for the suggestion. We have replaced the terminology ”subjects with HIV” with ”adults living with HIV (ALHIV)” in the manuscript.

Overall good well measured discussion.

Reply to General Comments: We thank the reviewer for the appreciation of our work.

Abstract: Page 1, Line 25: LV systolic ____. Word missing.

Reply to comment: We thank the reviewer for the observation. The word function has been added and now it reads as following:

            In the absence of cardiovascular risk factors, adults living with HIV display normal LV systolic function”

Introduction: Page 1 line 36: “rather” is wrong word choice.

Reply to comment: We thank the reviewer for the remark. We have now deleted the misplaced word. It now reads as following:

            The effective suppression of viral replication significantly lowered the prevalence of acute opportunistic infections and oncologic complications shifting the clinical spectrum of HIV infection towards chronic, more subtle conditions of the heart, liver and kidney.”

Page 1 lines 37-38:  references should be added to sentence “The relationship …”

Reply to comment: We thank the reviewer for the remark. We have now made the correction.

Page 2 lines 48-50: sentence “Moreover, …” Need to specific not in HIV population.

Reply to comment: We thank the reviewer for the suggestion. The sentence has been re-written as following:

Moreover, left atrial (LA) strain is an established predictor of future cardiac events in the general population [7].”

Page 2 line 64: Are you reported the gender or the sex?

Reply to comment: We thank the reviewer for the question. We have reported in the section study population that we have included 45 males in the study group and the control group was matched by gender. (see page 2 line 59).

Page 2 line 95: define PASP.

The abbreviation is  now explained in the text as following:

”Pulmonary arterial systolic pressure (PASP) [….]

 Results:

Page 5 Lines 168-170: Indentation error.

Reply to comment: We thank the reviewer for pointing out the error. This has been corrected in the text.

How was route of infection assessed? Is this accurate and relevant?

Reply to comment: Most of our patients (70%) acquired HIV parenterally (via IV injection with non-sterilized needles) during early childhood. This is the particularity of our study population. The route of infection was obtained from existing medical documents, the above-mentioned subjects having been diagnosed in early childhood with HIV infection.

Although the route of infection is not important for the overall impact of the disease, by selecting young adults with infection acquired during early childhood we could study the impact of HIV infection in subjects free from conventional cardiovascular risk factors. Conversely, most of the existing manuscripts addressing cardiac function in people with HIV included older subjects, with conventional cardiac risk factors and shorter duration of the disease.

Reviewer 3 Report

The manuscript by Oana Mirea et al. entitled “Echocardiography assessment of cardiac function in subjects with HIV: a speckle tracking study in the era of antiretroviral therapy” aimed to investigate whether subjects with HIV display latent cardiac dysfunction. The abstract summarizes the general significance of the manuscript and the article leads some evidence to such point, but the study sample is small and there are many cofounding variables. Consequently, some major issues need to be addressed to improve the significance of the manuscript:

-Firstly, the different classes of drugs taken by patients should be reported in Table1.

-Moreover, in exclusion criteria arterial hypertension and diabetes are not reported. Have patients with arterial hypertension or HIV-related pulmonary hypertension been excluded?

-Why did the authors use a univariate logistic regression analysis to identify the predictors of LA strain and not to identify the predictors of RV GLS and RV free wall strain? It would be interesting to know the result of this further analysis

- The bibliography is too limited. These articles should be cited:

-Athanasiadi E, Bonou M, Basoulis D, et al. Subclinical Left Ventricular Systolic Dysfunction in HIV Patients: Prevalence and Associations with Carotid Atherosclerosis and Increased Adiposity. J Clin Med. 2022;11(7):1804. Published 2022 Mar 24. doi:10.3390/jcm11071804

-Woldu B, Temu TM, Kirui N, et al. Diastolic dysfunction in people with HIV without known cardiovascular risk factors in Western Kenya. Open Heart. 2022;9(1):e001814. doi:10.1136/openhrt-2021-001814

Author Response

We would like to thank the editors and reviewers for the detailed review of the manuscript and their helpful comments. We have tried to address point by point the raised concerns.

Reviewer 3

The manuscript by Oana Mirea et al. entitled “Echocardiography assessment of cardiac function in subjects with HIV: a speckle tracking study in the era of antiretroviral therapy” aimed to investigate whether subjects with HIV display latent cardiac dysfunction. The abstract summarizes the general significance of the manuscript and the article leads some evidence to such point, but the study sample is small and there are many cofounding variables. Consequently, some major issues need to be addressed to improve the significance of the manuscript:

-Firstly, the different classes of drugs taken by patients should be reported in Table1.

Reply to comment 1. We thank the reviewer for the suggestion. We have added the treatment protocol in Table 1.

-Moreover, in exclusion criteria arterial hypertension and diabetes are not reported. Have patients with arterial hypertension or HIV-related pulmonary hypertension been excluded?

Reply to comment 2. Our study population included young adults without history of arterial hypertension and/or diabetes mellitus. We have indeed failed to mention the two criteria in the exclusion section. They are now added as following:

” Exclusion criteria were (1) history of cardiomyopathy, coronary artery disease, or valvular heart disease; (2) presence of diabetes mellitus; (3) history of arterial pressure above 140 mm Hg and (4) echocardiographic images unsuitable for quantification.”

-Why did the authors use a univariate logistic regression analysis to identify the predictors of LA strain and not to identify the predictors of RV GLS and RV free wall strain? It would be interesting to know the result of this further analysis

Reply to comment 3. We agree with the reviewer that such an analysis would bring additional value to the manuscript. We have now added the data in the results section.

 ”Using the univariate logistic regression analysis, parenteral infection (β = -0.527, p=0.032), duration of HIV infection (β = 0.711, p=0.004) and BMI (β = 0.404, p=0.004) were independent predictors of RV global strain. ”

- The bibliography is too limited. These articles should be cited:

Reply to comment 4. We thank the reviewer for the remark. The recommended papers are relevant for our study and are now addressed in the text and included in the reference section.

-Athanasiadi E, Bonou M, Basoulis D, et al. Subclinical Left Ventricular Systolic Dysfunction in HIV Patients: Prevalence and Associations with Carotid Atherosclerosis and Increased Adiposity. J Clin Med. 2022;11(7):1804. Published 2022 Mar 24. doi:10.3390/jcm11071804

-Woldu B, Temu TM, Kirui N, et al. Diastolic dysfunction in people with HIV without known cardiovascular risk factors in Western Kenya. Open Heart. 2022;9(1):e001814. doi:10.1136/openhrt-2021-001814

Round 2

Reviewer 3 Report

Thanks to the authors for the answers.